# Possibilities of Measuring and Detecting Defects of Forged Parts in Die Hot-Forging Processes

**DOI:** 10.3390/ma17010213

**Published:** 2023-12-30

**Authors:** Marek Hawryluk, Sławomir Polak, Marcin Rychlik, Łukasz Dudkiewicz, Jacek Borowski, Maciej Suliga

**Affiliations:** 1Department of Metal Forming, Welding and Metrology, Wroclaw University of Science and Technology, Wybrzeże Wyspiańskiego 27 Street, 50-370 Wroclaw, Poland; slawomir.polak@pwr.edu.pl (S.P.); marcin.rychlik@pwr.edu.pl (M.R.); lukasz.dudkiewicz@pwr.edu.pl (Ł.D.); 2Kuźnia Jawor, S.A, Kuziennicza 4 Street, 59-400 Jawor, Poland; 3Schraner Polska Sp. z o. o., Lotnicza 19G Street, 99-100 Leczyca, Poland; 4Łukasiewicz Research Network—Poznań Institute of Technology, Jana Pawła II 14 Street, 61-139 Poznań, Poland; jacek.borowski@pit.lukasiewicz.gov.pl; 5Faculty of Production Engineering and Materials Technology, Czestochowa University of Technology, 42-201 Czestochowa, Poland; maciej.suliga@pcz.pl

**Keywords:** measurement, defects, forging process, numerical modeling, 3D scanning

## Abstract

This paper presents research results in the field of industrial die forging, mostly related to the use of advanced measuring techniques and tools, numerical simulations, and other IT tools and methods for a geometrical analysis of the forged items as well as detection of forging flaws and their prevention, and optimization of the hot-forging processes. The results of the conducted investigations were divided into three main areas. The first area refers to the application of, e.g., optical scanners and programs related to their operation, data analysis, including the construction of virtual gauges, measurements of selected geometrical features of both the manufactured forgings and their physical and virtual models, as well as an analysis of the durability of the forging tools based on the proprietary reverse scanning method. The second area presents the results of measurements and analyses performed with the use of finite element modeling and by means of some special functions in the calculation packages, such as contact, flow lines, trap, or fold, for the detection of forging defects and an analysis of the force parameters. In turn, the third area presents a combination of different methods of measurement and analysis, both FEM and scanning, as well as other IT methods (physical modeling, image analysis, etc.) for the analysis of the geometry and defects of the forgings. The presented results point to the great potential of these types of tools and techniques in forging industry applications as they significantly shorten the time and increase the accuracy of the measurement, as well as providing a lot of valuable information, physical variables, and technological parameters that are difficult or impossible to determine either analytically or through experimental means. The use and development of these techniques and methods are fully justified, both in the aspect of science and the increased effectiveness and efficiency of production.

## 1. Introduction

The quality and dimension-shape precision of forgings are crucial as they dictate the usefulness of forged items for their dedicated applications. The correctness of these key features often also translates to some specific parameters or performance properties. This is a very important aspect for the recipients of the forged elements from the aircraft and military branches as well as the automotive industry, where the requirements for the precision and quality of the forged products are at a very high level. All this is related, on the one hand, to the proper control and measurement of the selected geometrical and quality features, which requires the application of the appropriate measurement methods and equipment. On the other hand, it should be noted that industrial hot forging is one of the most difficult-to-implement manufacturing processes due to the technological aspects and quality of the obtained products, as well as the durability of the forging equipment. This is because the shape of the forging is calculated with the consideration of the prices of the forging tools. It is necessary to produce the latter in a specific quantity and dimensional quality, with a constant number of forged elements obtained from a given punch or die. Therefore, with respect to the relatively well-mastered hot-die-forging technologies, the correct production of forgings with complex geometries that will meet the high dimensional and quality requirements of the consumer requires the extensive experience of the designers, technologists, operators, and blacksmiths [1]. It also creates the necessity to implement new forging projects, constantly improve the already realized technologies, and solve many issues influencing the success of the entire manufacturing process; their interactions make many forging processes hard to analyze or improve. The wear of the forging tools and other equipment results in changes in the geometry of the manufactured elements (forgings) and any flaws of the tools are reflected in the forging, reducing the quality and functionality of the final detail/product achieved from the forging. Faulty and incorrectly designed and damaged tools, as well as the technology itself, are the cause of many flaws in the items manufactured in the forging processes, i.e., the forgings [2]. The defects can be classified into three main groups [3]. The first group includes the defective shape of the forged products, i.e., incorrect geometry or dimensions, deformations, bends, skips, twists, underfills, folds, splitting, etc. [4,5]. The second group includes external and internal surface defects, such as laps, cracks, hot tears, overlaps, scales, an improper arrangement of the fibers, an improper grain size, etc. [6,7]. In turn, the third group contains defects connected with mechanical, chemical, physical, and technological properties [8]. Of all the forging defects, the ones most commonly observed are those connected with the shape (underfills and laps) and the surface, which result from an improper shape and/or arrangement of the slug forging/forging on the tools [9]. Such errors are a result of the complicated geometry of the designed forging parts, which makes it impossible to properly prepare the material or machinery park; in addition, this also affects the preparation of an ideal slug/preform for a proper filling of the working impression [10,11]. Moreover, sometimes, errors result from an incorrect charge diameter, which is caused by the lack of a specific bar profile in the steelworks ([7], also [12,13]). Other causes of defects in the forged products include an improperly selected temperature of the input material and the tool, excessive bending and deformation, a poor design of the tool blanks, improper lubrication, and failure to remove the scale [14]. A large part of these defects depends on the proper operation of the die forging; however, discrepancies often occur for reasons beyond the control of the company, and they should be controlled and supervised in order to stop the deterioration of the quality and dimension-shape accuracy of complicated forging items [15,16]. For this reason, at present, more and more often, for the analysis and measurement of the selected quantities, and also the optimization of the whole forging process, classic or more advanced measurement tools are used, such as laser scanners, as well as a range of IT tools [17], often aided with numerical methods based on FEM/FVM [18,19,20].

The previously used analytical methods, usually based on a visual assessment or a measurement of the crucial geometric features performed by means of traditional contact methods and tools or special measurement tools, are being replaced with non-contact measurement methods based on spatial scanning [21]. Additionally, measurement methods are constantly being improved [22,23], e.g., through the application of measuring arms, special probes, or trackers, which make it possible to measure large-size items [24]. This makes such devices ideal for the assessment and comparison of point clouds with CAD models, as well as for reverse engineering, rapid prototyping [25], spatial modeling, fast prototyping, and 3D modeling [26,27]. Despite their lower accuracy, 3D scanning devices can be an alternative for CMM [28,29]. Measuring arms, owing to their special mobility, allow for the movement of the measuring station between places within the production lines in the company (e.g., from the labs to the forging process) in a short time, directly to the examined item (near the forge), in order to perform a quick measurement and analysis of the results [30]. For example, measuring arms with integrated scanners fulfill similar functions, such as assessment of the quality of the workmanship, diagnostics of the defects of large-size items reforged on hammer aggregates, and analysis and evaluation of relatively small forgings on the production line [31,32]. However, in the case of surface quality measurements of the forgings, profilometers and roughness meters are used. In turn, when the forgings do not characterize in good parameters (in the case of huge sizes), 3D scanners are used as, in addition to measuring the geometry, they enable simplified measurements of the surface waviness to be performed when it is important to quickly obtain an image of the condition of the entire tool [33].

On the other hand, a more advanced tool used to analyze and measure as well as detect defects in forged elements [34,35] is FE modeling, which belongs to non-destructive tests. In a sense, using numerical simulations based on FEM, it is possible to make “virtual” measurements of selected geometrical features, both of the forging tools and the forgings and also even more advanced multi-variant analyses [36,37]. Furthermore, a combination of numerical modeling and other measuring techniques, e.g., 3D scanning, microstructural tests (destructive), and thermovision measurements, brings measurable advantages resulting from a quick verification of the results obtained. This translates to measurable economic aspects related to the production process and the control of its correctness (including the obtained products) [38,39]. It should be emphasized that most of the key technological parameters and physical quantities can be determined based on FE modeling through a virtual experiment, as the currently used calculation packages/programs [40] make it possible to determine other technological parameters that are difficult to find analytically [41,42]. The currently applied FEM programs have many functions allowing for an even more accurate and compressive analysis of metal-forming processes. They make it possible to not only determine the distributions of plastic deformations or temperature fields but also the force-energy parameters and good filling of the tool’s working cavity, i.e., the correctness of the representation of the forging geometry, as well as to detect defects in the forged items [43]. The application of such functions as contact, laps, folds, or flow lines, as well as other typical program functions, significantly reduces the realization time of a new project and limits the errors in the tool design [44]. The available literature provides studies whose authors use many of the mentioned methods and tools simultaneously, where the results point to high effectiveness in solving many difficult issues connected with hot-die-forging processes [45] and where FF simulations are verified in real processes [46]. Therefore, this approach, consisting of combining the presented methods and measurement techniques, is the right direction in the development of scientific and research work, both in solving problems in the field of production technology and materials engineering and the industrial implementation of the developed solutions [47,48]. The novelty of the presented work and investigations is demonstrated by the usefulness and practicality of the IT tools in the form of scanning and numerical modeling in applications in the forging industry, where, due to difficult operating conditions and certain historical background and financial problems, it is difficult to implement this type of methods as tools supporting the everyday work of the technologists and engineers. Also new is the use of the 3D reverse scanning method developed by the authors to assess the condition of the forging tools based on forging measurements (without the need to interrupt the process). Similarly, the use of special functions in computational packages and their development for the analysis of the forging processes, along with the verification of the performed simulations, is a new and practical approach to a virtual experiment because it is cheaper than a real experiment. This approach and the use of the presented methods allow for a thorough and relatively quick analysis of the problem as well as verification of the developed solutions without the unnecessary, time-consuming, and expensive tests in the real industrial process.

## 2. Materials and Methods

The aim of the work was to present many of the authors’ own investigations related to the measurements of selected geometrical features of forged products and ways of detecting defects in forgings manufactured in hot-die-forging processes, as well as the prevention methods, mainly with the use of advanced measurement techniques based on 3D scanning and numerical modeling.

The measurements of selected geometrical features of forged products are presently realized by means of classic measuring devices (slide calipers, micrometers, thickness gauges, etc.) as well as more advanced devices, such as CMM machines and 3D scanners. Currently, the second group is very popular, serving to perform precise scans and measurements. Spatial (3D) scanning can be realized by means of many different techniques, but it should be remembered that each of them has strengths and limitations. In hot-die-forging processes, measurement by non-contact scanning is used for two groups of objects: measurements for the evaluation of the forging equipment and measurements of the key geometric features of the forgings, as well as quality control of the resulting final detail [49,50].

Defects in the form of laps are the most difficult to diagnose, and they usually occur during an improper flow, where a part of the material remains between the dies, forming folds, which, in the consecutive strikes/operations, is pressed into the flash, but often also into the forging, which is unacceptable and completely disqualifies such products.

Of course, there are also other defects besides the ones connected with the required dimension-shape accuracy, underfills, and laps. In turn, at present, to conduct an analysis of the material flow and the occurrence of possible forging defects, numerical modeling is used, which, by means of more or less advanced functions, makes it possible, with high accuracy and probability, to detect or diagnose a defect in the forging process.

In the conducted research studies, the following main methods/tests and measurement devices were used:-Scanning the forging parts and tools with the use of a 3D scanner RS2 (ROMER ARM 7620si (Hexagon Manufacturing Intelligence, Aarau, Switzerland) together with the PolyWorks software 2017). The accuracy of the Absolute ARM 7520si according to the norm B89.4.22 equals 0.053 mm. This device makes it possible to collect over 460,000 points/s for 4600 points on the line with the linear frequency of 100 Hz;-Numerical modeling by means of the Forge 3.0 NxT package (Transvalor, Biot, France) and Marc Mentat;-Microstructural tests performed with the use of a light microscope Olympus BX52N (Tokyo, Japan) and a stereoscopic microscope Keyence VHX-R400E (Osaka, Japan);-Other classic measurement and research methods.

## 3. Results and Discussion

### 3.1. Measurements of the Forging Geometry

An example of such a measurement, presented in Figure 1, is the use of a technique of controlling the key shape/geometry features of the profile and/or surfaces of the forging on virtual fixtures, with special measurement databases developed, e.g., in the PolyWorks environment [21].

In the process of forging quality assessment and control, we can encounter successful measurement applications using scanners of various types as well as mobile devices, which can work almost under any conditions.

In the case of a geometry analysis, we can use 3D scanning for a geometry analysis and measurement of a Pb forging obtained as a result of tests performed through numerical modeling of an industrial process of manufacturing, such as forging from steel and aluminum as target materials (Figure 2a). Similarly, scanning can be applied for the analysis of the changes in the shape of the particular forged items from the target forging (Figure 2b). Also, analyses can be performed for multiple systems, both of particular forgings in the leaf (Figure 2c) and the possible bends of the whole leaf—the forged element (Figure 2d).

In this case, we can easily and relatively quickly verify in which areas (Figure 2a) the forging from the physical modeling is within the dimensional tolerance field (±0.2 mm for the physical model) and where changes could be made in the tool geometry so that the correctness of the product can agree with the specification. Also, for the other presented scanning results, verifying analyses can be made, for example, whether the dimension-shape accuracies of single forgings (Figure 2b) are in agreement with the tolerance, as well as, in the case of larger forged elements (multiple systems) [47] or whether the bend of such an element is acceptable, e.g., in the aspect of the trimming process (Figure 2c,d).

Using 3D scanning, we can easily measure and determine the so-called joggle, i.e., the difference between the lower and upper part/surface of the forgings described in reference to the parting plane. In Figure 3a, we can see a compilation of results referring to the twisting defect for the entire forging consisting of two forged parts in a double forging system [20].

The investigations performed in this field were related to analyses of the forgings’ joggle on the reconstructed crank press, where, based on the examinations, incorrectness of the mounting of the upper and lower tool set was established (incorrect referencing on the taper keys and twisting of both tools in respect of each other). By the designation of the resultant of the orientation deviation at the moment when the press slide is in the bottom turning point, composed of a joggle member with the X and Y variable, it is possible, analogically, to determine the joggle value along axis Y to the case described for the value along direction X.

By means of the GOM Inspect Professional program, we can use the 3D scanning data for a comparison of the introduced changes as well as some more improvements or optimization of the process with respect to the technologies implemented so far. For example, Figure 4 illustrates the dimensional deviations of the produced forging: a fork for the cardan shaft in excavators (Figure 4a), adapter forgings (Figure 4b), or forgings of a hub for the power transmission in a motor truck gearbox (Figure 4c) from the nominal dimensions before and after the introduced changes. On this basis, we can conclude whether the introduced changes, e.g., a redesigned tool or changed tribological conditions, have brought the desired effect of improvement of the dimension-shape accuracy for the analyzed forged parts.

On the basis of the presented comparison of the scans (before and after), we can conclude that, as a result of the introduced corrections into the industrial process, more accurate products were obtained, i.e., forgings with a narrowed allowance field for the dimensional tolerance, which justifies the use of FEM for the optimization of the forging processes. By analyzing the results obtained by way of comparing the 3D scans (before and after the changes), it can be observed that the improvements introduced into the entire forging process made it possible to obtain more accurate products—forgings with a narrowed dimensional tolerance area. This allows us to conclude that the use of numerical simulations to optimize the forging processes is fully justified. Figure 5a shows another example of applying the results of 3D scanning performed by means of a laser scanner to analyze the geometric changes in a selected surface/area of disc-type and lid-type forgings (Figure 5b) periodically collected from the process. The results for the subsequent forgings presented in Figure 5 indicate progressive wear of the forging tools (this should be understood as a geometric loss of the tool material). This material loss is located on the circumference and the central part in the case of a disc-type forging and the case of a cover-type forging (in its front part)—in the vicinity of the ejector hole. Moreover, it can be observed that, for both forgings, in the initial phase of the forging process, such wear is irregular. In turn, with the increasing operation time in these areas, we can notice increased wear as well as the appearance of new areas, proving the progressing wear [21].

The obtained 3D scan results in the form of a shape lapse of the cyclically collected forgings enable a relatively simplified analysis. Owing to the development of the reverse 3D scanning method (Figure 6a,b), it is possible, through a measurement (scanning) of the forgings collected from the process, to describe the operation history of the forging tools that made these forgings [48]. Figure 6c illustrates a comparison of the geometrical and dimensional changes in the forgings after a certain number of them have been made and the wear of the filler (corresponding to the closer number of forgings in the model and at the end of its operation time). Figure 6d shows a diagram of the geometrical changes—a material loss on the selected surface based on the volume change tests (adequately to the growth) of the forgings.

Based on the presented diagrams (Figure 6b,d), which, with their character, resemble the so-called Lorentz wear curve, we can observe the relations and distinguish between the three stages of wear. The achieved results of a shape lapse of the cyclically collected forging items allow us to determine the areas in the tool where a loss or a growth of the material volume takes place. In the case of an analysis of the data for the cyclically collected forgings, the material growth in the forging means a loss (wear) in the tool, and vice versa. Such a recreation of the wear enables an analysis conducted at an interval equaling the frequency of the forging collection, i.e., elaboration of a wear curve for the tool based on the forgings. It should be clearly emphasized that the last measured forging is verified with the measurement and determination of the changes in the volume of the tool for the maximum number of produced forgings (i.e., the 12,000th forging out of the 12,500 pieces produced by the tool). The differences in the initial period can probably be caused by the process stabilization (the entire forging system), i.e., with the assumption of the proper tool work temperature, i.e., the tribological conditions. The big differences at the final stage of wear time can result from the fact that, for this range, the loss of the tool material is larger than the material growth on the forging.

This has so far been very difficult to realize only for the tools due to the interruption and interference in the production process. In turn, the comparisons of both methods shown in Figure 6d, on the example of a disk forging (interrupting the production and scanning the tools during their work and scanning the cyclically collected forgings), point to a big agreement. Moreover, with the use of scanning, it is possible to additionally divide the forging surface into several characteristic areas, in which separate analyses can be carried out [20]. Analyzing in detail Figure 7, we can observe that, on a few forgings, in their external part (circumference), areas of material loss/lack (negative volume) are present, which probably proves the occurrence of a defect in the form of an underfill. In turn, based on the scan results of the forgings (positive volume—growth of the forgings material), wear can be observed in the central part of the tool, in the vicinity of the ejector hole on the front section of the forgings, which, at the beginning of the process, is irregular. A slightly different situation is observed in the radius, on the exit from the die (the deepest section of the die insert), where we can notice clear shape lapses, pointing to an underfill of the forging. In turn, the obtained results point to a loss of the tool material in the area right before the bridge and the transition into the flash as an asymmetric ring, which suggests tool wear. Therefore, the forging can be divided into 3 characteristic areas: A, B, and C, in which volume growths and losses will occur (positive and negative values on the forging constituting a converse on the tool). Next, on this basis, we can perform a global evaluation of the process, which is the quality of the forging, the tool life, as well as other aspects of the production processes. Then, based on the analysis of the volume changes in the forging material, a global wear curve was drawn (Figure 7b).

Areas A and C are dominated by a “positive” volume of the forged items, which proves intensive die wear. Even though both zones have similar surface areas, area A is more heavily worn. However, in area B, in the case of periodically collected, scanned, and measured forgings, a negative volume is dominant. In the analyzed surface of area B, which is three times larger than the other areas, the visible changes are relatively small, both in terms of tool wear and the quality of the forgings themselves. The single “jumps” in the value of the “negative” volume of the consecutive forgings visible in the 3D scan results in Figure 7b are concerning, and they suggest even more the presence of forging flaws in the form of underfills and lack of process stability. A cause of this can be the lack of good repeatability of the optimal friction conditions resulting from the insufficient vaporization of the liquid from the lubricant dose. Other reasons may be the scale temporarily sticking to the tool, e.g., during technological breaks. In effect, the formed forging material is not able to fill the impression in some areas. This is a preliminary die forging operation, and the analysis of the geometric correctness of the forgings after the finishing stage did not reveal any significant defects in the forging, resulting in material deficiencies. At the same time, because, in area B, slight changes take place, and the area can be treated as showing small wear, it can be a base area for aligning the data from the point of view of the selection of the mating surface in the reverse method. However, we should clearly emphasize that measurements of the quality and geometry of the forgings in hot-die-forging processes are very difficult, and it is hard to point to an optimal solution, as optical measurements, due to the high temperature and its dynamic changes in reference to the emissivity, will be burdened with a big error. It should be emphasized that the measurement itself of the forgings cyclically collected from the process lasts about 1 min, and it should be considered that, directly after the forging process, the forgings’ temperature equals about 1000 °C and it begins to drop at an average rate of (depending on the forging’s size) about 100–150 °C per minute. Of course, the lower the temperature, the slower the cooling. On the other hand, the lower the temperature, the lower the thermal expansion and the smaller the scale. Therefore, we can assume that, after about 10 mins, the measurement performed through scanning should not be burdened with a big error (with the assumption that the forging is clean of the scale).

The proposed approach based on the reverse scanning method seems a relatively good solution as, when it is combined with fast processing of the obtained data and its handling, we can obtain in a short time, e.g., important information about the process in the form of notifications sent to the phone about the state of the forging quality in relation to the state of the tools as well as the whole process.

### 3.2. Detection and Analysis of Forgings Defects

#### 3.2.1. Application of the Flow Lines/Surface Function

A quite popular function in numerical modeling enabling an analysis of the flowing manner involves the use of the so-called “flow lines”, which are introduced at the beginning of the modeling process in the geometry of the feed material in the form of lines, e.g., in the longitudinal and/or transverse direction. Figure 8 presents the results of the material flow lines, which is very important in the aspect of possible improperness, which may lead to laps or insufficiently good performance properties as a result of anisotropy.

A thorough analysis using the modeling method based on FEM was conducted of the flowing manner of the deformed material on a geometrically complicated forging of a motorcycle transmission power lever with the use of the flow lines along both directions (Figure 9).

The results of the FE modeling showing the fiber distribution in the forging in two directions confirm that flaws in the form of overlaps can appear in the analyzed areas, especially in area 2, where a deep pin/joggle is visible, on which we can see that the fibers, in both directions, have a great tendency to intersect and fold.

The case is similar to the results of the sub-surface lines for a forked forging forged in a double system. Figure 10a shows the course of the lines in the vicinity of the parting plane of the dies. The continuity from the forging’s arm through the rear section is maintained without clear intersections of the particular elements. Figure 10b presents a similar correct tendency for the sub-surface lines arranged in the arm of the forging. Often, such analyses can be combined with the function of the Garfield factor, according to which the value of a parameter presenting a risk of a defect is >0.7.

Figure 10c shows the area of the occurrence of the highest Garfield factor value in the zone of material deficiency on the flash (1.8 mm before reaching the lower forging position). This results from the flow of the flash on both sides and their contact in the marked area. In turn, at the end of the forming process (Figure 10d), we can see no risk in the indicated area, and the highest values (0.44) occur on the arms, which do not cause a risk of defect.

#### 3.2.2. Application of the Contact-Type Function

For the analysis of the properness of the deformed material’s flow and degree of filling of the tool’s working impression, contact detection is often used in FEM. Figure 11 shows the numerical modeling results for a 3-operation process of reforging a needle rail type 60E1A into a rail profile 60E1.

The obtained results point to those areas where the material did not flow or flew improperly. They are mostly the areas marked with red and yellow, where the distances of the deformed material from the contact with the impression surface reach 2 mm. The areas on the left (both top figures) are places where there was no deformation of the tool (hence their red color). Figure 11b presents a magnification of the areas of the reforged end, where we can clearly see that the material did not fill the impression. However, the underfill visible on the end of the reforging of a longer needle rail does not constitute a problem, as it is contained within the cut-off part, 100 mm long, and in the industrial process, this end of the material is cut off at the length of about 100–150 mm. In turn, a problem is encountered, especially with the bottom part of the rail head, where such an underfill is unacceptable. In their study [21], the authors performed multi-variant numerical modeling preceded by physical modeling from Pb of this process, presenting a way to solve this problem.

The function of contact detection between the deformed material and the tool can also be used in combination with other quantities or physical parameters, e.g., the distribution of plastic deformation and temperature field or the course of the forging forces, to perform an in-depth analysis. Figure 12 shows such an analysis for a forked-type forging forged in a double system.

The results presented in Figure 12 enable a complex analysis of the entire forging technology, together with the force parameters, in the context of the possibilities of the occurrence of potential defects or other errors, both in the forging and in the whole process, with the purpose of its optimization.

Using the “contact” function, you can perform analyses determining which parts of the deformed material are still flowing (the material is deformed) and which no longer are, and you can also stop the virtual process at a given moment to assess whether the force parameters do not exceed the assumed ones. Moreover, such an analysis allows us to predict whether the deformed material flows as expected (in terms of fiber distribution) or whether corrections are necessary for the form of changes in the geometry of the shaping tools.

#### 3.2.3. Applications of the Trap Function-Air Pockets

A further and even more advanced step of analyses with the use of FEM can be the use of the trap function, which enables the prediction of the so-called air pockets. For an analysis of the underfilling of the impressions in the vicinity of the “pins”, further FE simulations were conducted, and the way in which those areas were filled was analyzed (the contact function), as well as some new functions in the FE programs were applied: “trapped surface”, “trapped volume” and “trapped pressure” for the analysis of the problem of pin filling in the forged element with single forgings (Figure 13). Based on the obtained results, underfills can be noticed, which show the size of the surface that remained unfilled and trapped, preventing further flow of the material (Figure 13a,b). Moreover, the analysis shows that the largest trapped surface can be observed in the vicinity of the pins, which may result in problems with filling these areas (Figure 13c). In the case of pressure distributions in these critical areas, they are much higher than in the remaining ones and amount to approximately 2000 MPa (Figure 13d).

Such values significantly impede the flow of the formed forging material. The reason for this may be both the trapped air creating “air pockets” and the lubricant residues in these areas, which at the same time constitute a barrier for the proper flow of the deformed material and their complete filling.

#### 3.2.4. Application of the Folds Function—Folds and Laps of Material

Defects in forged products can also be identified with the use of the fold function, where folds are presented in the program post-processor in the form of a cloud of red dots. Figure 14a,b presents the formation of a lap in a yoke-type forging, especially in the area of the arm and the movement of the material into the flash—in this case, it is not dangerous for a forging in the industrial process, as the defect is in the flash. In turn, Figure 14c,d presents the subsequent steps in the formation and development of the fold in the tested element, revealed owing to this fold function.

In turn, Figure 14d shows that reducing the distance from the impression’s edge (the charge moved closer) assured that, for the same element, based on the FE modeling, no defects in the form of folds or laps were observed.

The use of the special folds functions allows us to simulate various variants of the forging position and then detect those areas of the deformed material that are at the highest risk of defects. The analysis showed that, during the flow, a part of the material remains between the tools, creating a fold (Figure 15a), which, in subsequent impacts, is pressed into the flash. During the finishing forging, there is a continuous tendency to outflow beyond the detail, which prevents the defects from affecting the product’s parameters. In the analyzed process, small defects can be formed in the corners in the elongation process on an anvil, as it is a lap-generating process (Figure 15b).

The application of the folds function is very useful in the analysis of defects of this type, as it makes it possible to easily and quickly identify that area in the forging where, with high probability, we can observe the occurrence of a defect in the industrial process.

A similar situation with defects in the form of laps was observed in the industrial process of a very important hub-type element assigned to power transmission systems in motor cars (Figure 16). Flaws in the form of grooves or laps were observed (red arrows in Figure 16a), which can also be caused by an improper geometry of the tool, which is shown by the “folds” function in FE modeling (Figure 16b). In the process, sometimes, another defect was detected, in the form of a lap, which was caused by an incorrect position of the set of tools: upper punch to the lower die (Figure 16c,d).

Defects of this type result from inadequately designed working cavities of both inserts. The use of the laps function makes it possible to predict the occurrence of such defects and to take the proper remedial measures.

The application of the presented functions (e.g., contact, trap, laps, Garfield coef., etc.) in the FEM programs enables a simulation of different variants of arrangement, as well as detection of those areas in the deformed forgings material which are the most exposed to flaws.

#### 3.2.5. Combination of Numerical Modeling and Measurements of Selected Geometrical Features of the Forgings

With the use of the modeling results and measuring techniques based on, e.g., 3D scanning and analysis, we can perform a mutual verification of the FEM model and the geometry of the forgings obtained from the industrial process. Figure 17 presents a comparative analysis of both the shape and dimensions of the 60E1A6 rail determined based on the numerical simulations in relation to the developed nominal value—the CAD model [21].

The results of the comparison show a good agreement of numerical modeling with the assumed CAD model of a reformed rail. The biggest differences are in the areas where allowances have been provided before the machining treatment.

Figure 18a shows the results of tests of a production process for partially worn tools used forging and trimming as a comparison of the results of computer simulations after truncation. In turn, Figure 18b shows the results of scanning for a forging obtained from an industrial process to nominal dimensions [21].

The analysis of the results allows for the conclusion that both the shape and deviations in the trimming zone of the forging from the simulation and the forging obtained in the industrial process are consistent. For the numerical model, the width of the cutting line is in the range of 2.1 to 3.1 mm, and the deviations are 0.1 to 0.3 mm. The obtained results are similar to a forging obtained in an industrial process, where both the width (2.1 to 3.4 mm) and the deviations from the nominal direction are slightly larger and more diverse along the entire cutting line (0.12 to 0.26 mm). It should be emphasized that, for the numerical model, such a process is carried out under “ideal” virtual process conditions.

Within the research studies, tests were also conducted referring to the optimization of the forward extrusion with the use of physical and numerical modeling. The numerical thermodynamic model of the actual process was built in the Forge software [36]. In computer simulations of reinforced aluminum extrusion, all tools, i.e., the die, the punch, and the container, were assumed to be rigid, with a heat exchange. In the case of boundary conditions, the assumptions were closest to the conditions occurring in the real process. The Coulomb friction model was used, assuming a value of 0.06 for all the elements. During extrusion, both in the real process and the physical modeling (Figure 19), the formation of “dead zones” in the corners of the die was observed. Only after a thorough analysis was the value of the friction coefficient for individual elements changed. In the case of the punch, the assumed friction coefficient was 0.05, whereas for the container, it was −0.12, and for the forging die, it was −0.27. Figure 19b presents the force courses for the real process and the FE modeling with new tribological conditions.

The introduction of new values of the friction coefficient and other initial boundary conditions resulted in the obtained results being very close to the real process. Some differences were observed in the level of the force, which, for the modeling, was approximately 5% lower compared to the real process. This may be due to the assumption of a flat deformation state in FEM, which meant that the numerical model did not take into account the friction between the container and the side wall of the molded part. A decision was made to carry out a complete analysis of the similarity of the flow of the deformed material and the distribution of strains using the ASAME digital analysis program to compare the results obtained from FEM and the real process (Figure 20a).

Similar tests were performed for the improvement of the backward extrusion of Pb and its corresponding physical model with the use of synthetic wax filia (Figure 20b).

As shown by the research and R&D work carried out, it is necessary to verify the boundary conditions when designing and analyzing plastic forming processes. This can be done with the use of a relatively expensive industrial process as well as fast and inexpensive physical modeling. Moreover, it seems that the best solution is the simultaneous use of numerical and physical modeling as mutually verifying tools for analysis and improvement of the industrial process.

## 4. Summary

The article presents the results of many years of research studies referring to the possibilities of using a whole spectrum of modern measuring techniques and tools as well as numerical simulations for a geometrical analysis of forged elements and the detection of forging defects together with the ways of their prevention, as well as optimization of the hot-forging process. The results of these investigations were divided into three main parts, including an analysis of the process of die forging at elevated temperatures. The first area referred to tests connected with the use of 3D scanners together with the application of software for complex data analysis, including, e.g., construction of virtual gauges and measurement of selected geometrical features of both the forged products and their physical and virtual models obtained from FEM. Also, the possibilities of the application of the proprietary method of reverse scanning were presented for the analysis of the forging tool durability, with the possibility of dividing the whole tool into key parts and performing advanced analyses of the whole production process. The proposed approach based on the reverse scanning method seems a relatively good direction, as, when combined with quick data processing, followed by data handling, we can obtain in a short time the crucial information about the process in the form of phone notifications about the state of the forging quality in reference to the state of the tool as well as the whole process. In the second part, the unlimited potential of numerical modeling was presented, used for the measurements and analysis of the process properness, the force parameters as well as the forging quality, owing to the use of basic functions, such as contact, flow lines, and also some more advanced ones, e.g., Garfield criterion, trap or folds, for the detection and prediction of forging defects. The last part presents the possibilities of combining different measurement and analysis methods, both FEM and scanning, as well as other IT tools (physical modeling, image analysis, etc.), for the analysis of selected geometrical features as well as forging defects.

The application and development of the described techniques and methods of measurement and simulation are fully justified, both in the scientific and economic aspects, because the presented results point to big potential for tools and techniques of this type in the forging industry, as they significantly eliminate the real experiment and shorten the time of analysis as well as increase the measurement precision, at the same time, providing much valuable information about the industrial process and the technological parameters as well as some physical properties which are difficult to determine in another way.

## Figures and Tables

**Figure 1 materials-17-00213-f001:**
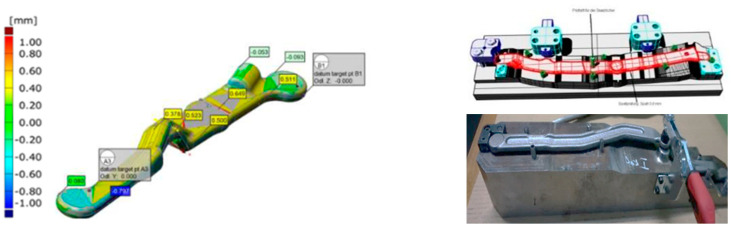
Examples of virtual gauges for forged product control.

**Figure 2 materials-17-00213-f002:**
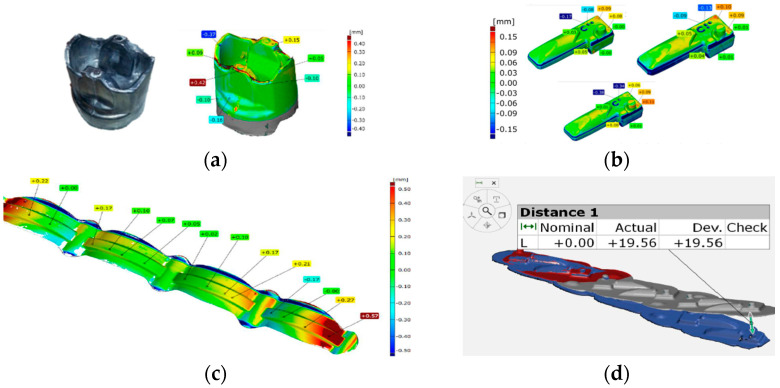
Results of 3D scanning analyses: (**a**) a comparison of the results for a Pb piston forging from physical modeling with the CAD nominal model, (**b**) a map of dimensional deviations for three single forgings after trimming from a larger element, (**c**) an analysis of deviations for clamping ring forgings and (**d**) bending forging part in multiple systems [21].

**Figure 3 materials-17-00213-f003:**
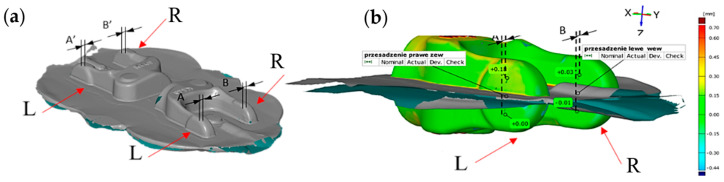
The view of the results of the defect analysis in a double system: (**a**) the entire forging item, (**b**) a half of the forged part, presented on a colored map of deviations; L-left, P-right side.

**Figure 4 materials-17-00213-f004:**
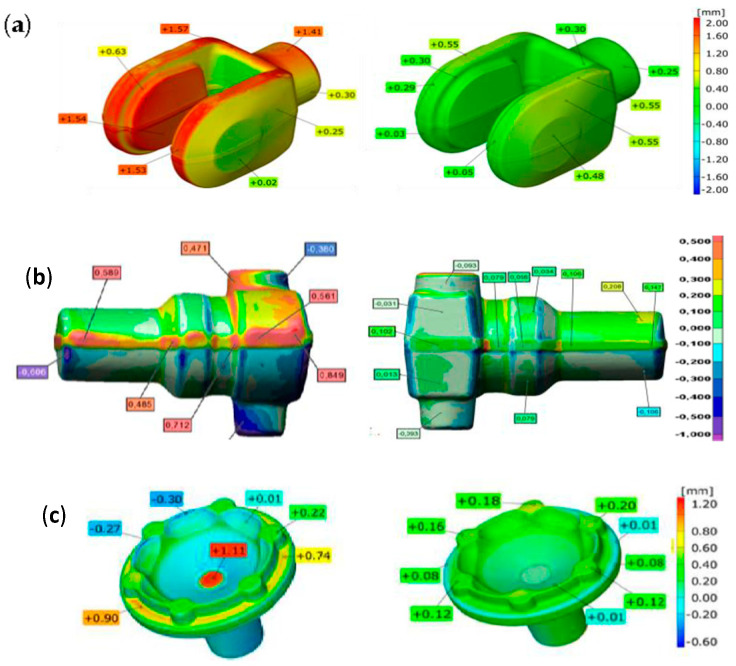
Comparison of the manufactured forging with the nominal: (**a**) a fork for the cardan shaft in excavators, (**b**) forging of adapter, (**c**) forgings of a hub for the power transmission; left side—before the changes, right side—after the changes from FEM [21].

**Figure 5 materials-17-00213-f005:**
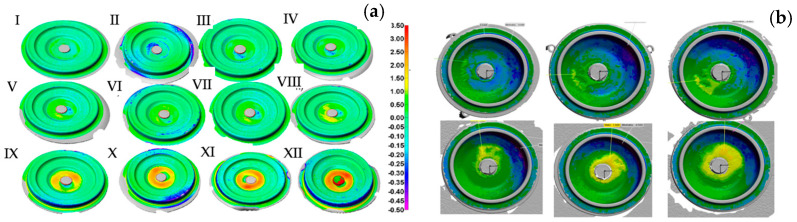
Scanning results for cyclically collected forgings of: (**a**) a spur gear disk from 1000 to 12,500 items (every 1000th item), (**b**) a lid from 2000 to 12,000 items (every 2000th item).

**Figure 6 materials-17-00213-f006:**
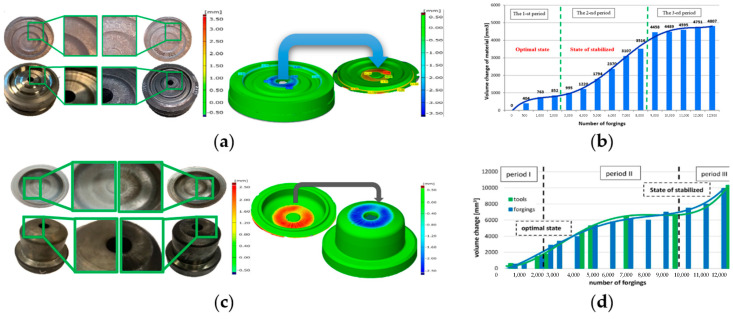
The concept of the reverse 3D scanning method: (**a**) the idea for a disc-type forging, (**b**) a lid-type forging, (**c**) a diagram of the volume changes based on the scanning results for the lid forgings, (**d**) a comparison of the scan results for the forged parts and the die inserts for a process of disc-type forging.

**Figure 7 materials-17-00213-f007:**
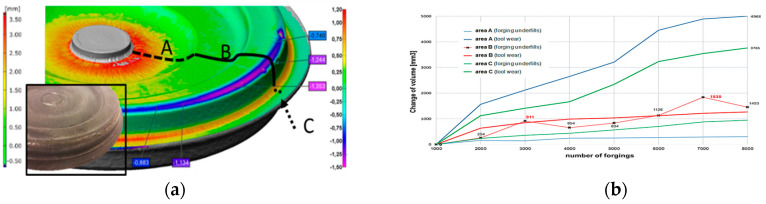
The idea of the 3D scanning reverse method for a disc-type forging: (**a**) a division of the selected forging surface into 3 characteristic areas, (**b**) a volumetric consumption chart for the increasing number of forgings in the process for the 3 selected areas.

**Figure 8 materials-17-00213-f008:**
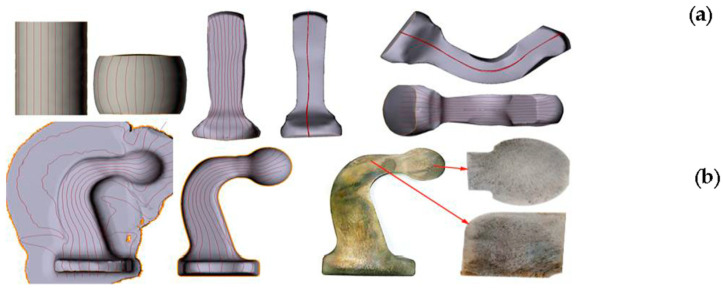
Analysis of the properness of the flow for a process of manufacturing a tow hook forging: (**a**) the flow lines from selected operations obtained from numerical modeling, (**b**) the fiber distributions obtained in a Jacewicz test.

**Figure 9 materials-17-00213-f009:**
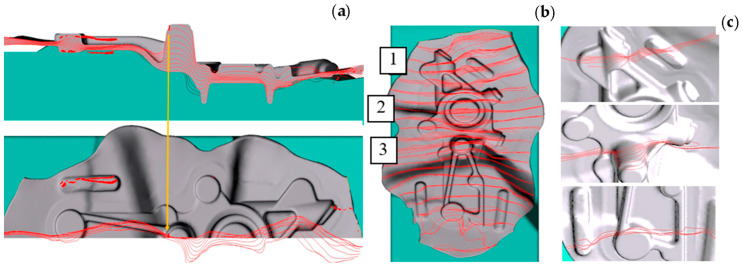
Analysis of the flow and the fiber distribution from the simulations: (**a**) photos of the lines arranged in the longitudinal direction, (**b**) in the transverse direction, (**c**) detailed analysis for selected areas from Figure 6b.

**Figure 10 materials-17-00213-f010:**
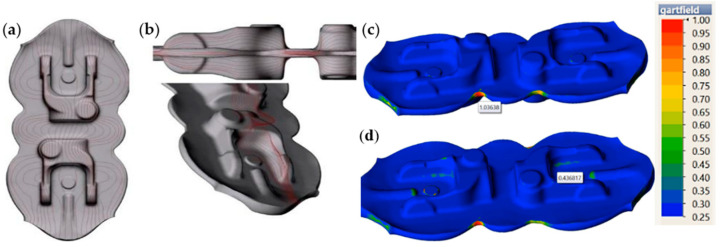
Analysis of the sub-surface lines of a forging during the roughing operation in a double system: (**a**) the lines in the vicinity of the parting plane, (**b**) the lines in the plane of the forging’s arm, (**c**) the Garfield factor value of a forging during roughing in a double system: (**a**) 1.8 mm before the end of the process, (**d**) at the end of the forming process.

**Figure 11 materials-17-00213-f011:**
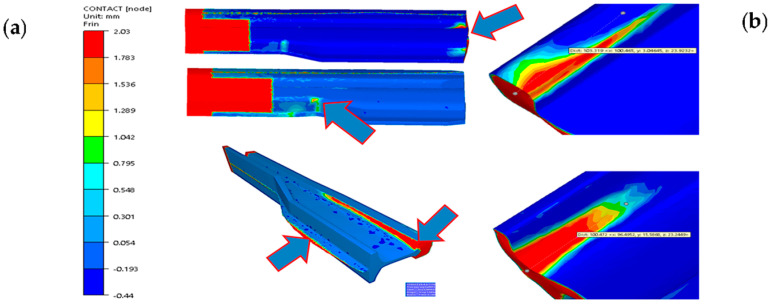
FEM analysis of the contact: (**a**) in the last deformation step in operation 3, (**b**) magnification of the areas with underfills in the head and on the foot [21].

**Figure 12 materials-17-00213-f012:**
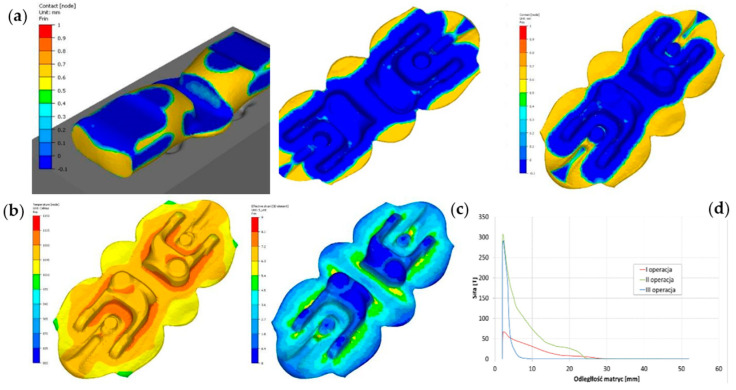
Results of a FEM simulation for a yoke-type forging in a double system: (**a**) the contact in the form of the distance of the deformed material from the impression for the consecutive operations, (**b**) the distributions of temperature in the finishing operation, (**c**) the plastic deformation distributions for the finishing forging operation, (**d**) the forging forces for the particular operations.

**Figure 13 materials-17-00213-f013:**
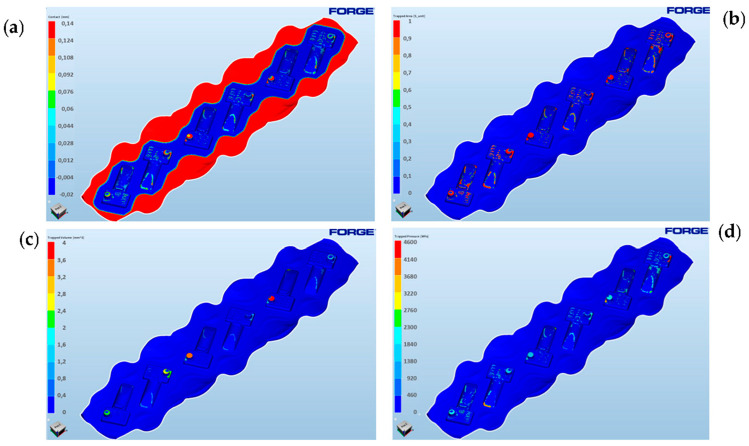
The FE modeling results: (**a**) filling of the pins’ cavities—contact at the end of the deformation process, (**b**) the trapped areas in the pins, (**c**) the underfilled “free” volumes, (**d**) the distribution of the pressures towards the end of the deformation [22].

**Figure 14 materials-17-00213-f014:**
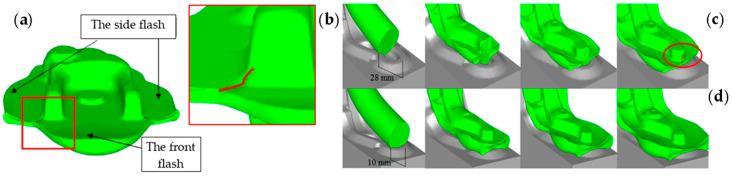
The FEM results obtained by means of the functions “folds”: (**a**) visible laps in the forging’s arm transitioning into the flash, (**b**) enlargement of the area with the defect, (**c**) the charge material moved 28 mm from die insert end, which caused a fold at the end of the forging process, (**d**) the input material moved 10 mm away—there are no folds at the end of the process [20].

**Figure 15 materials-17-00213-f015:**
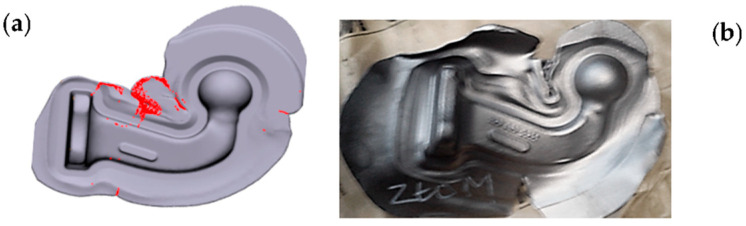
Forging defects: (**a**) a fold on a forging during roughing and (**b**) defects on a forging after finishing forging made from a too-long slug forging.

**Figure 16 materials-17-00213-f016:**
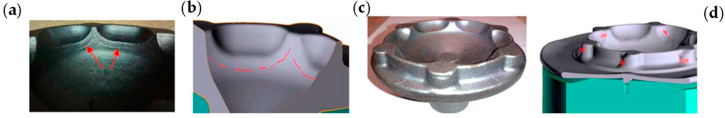
Exemplary results of the detection of forging defects of a hub-type forging: (**a**) laps and reflections in a forging, (**b**) FE modeling with folds, (**c**) laps on the front surface, (**d**) FEM—rotational displacement of the tools in their axis [21].

**Figure 17 materials-17-00213-f017:**
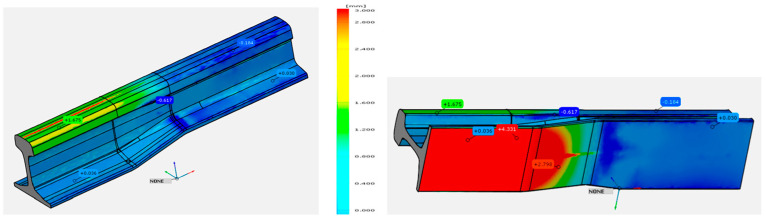
Distribution of shape calculated from FEM from the nominal shape and dimensions from the CAD model.

**Figure 18 materials-17-00213-f018:**
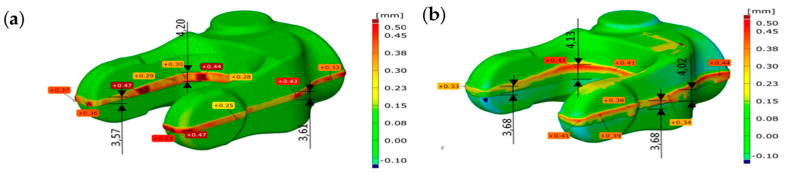
A comparison of a forging model after trimming of the flash: (**a**) obtained from FEM with the nominal CAD model, (**b**) obtained from an industrial process for similar conditions (a scan of a forging with the nominal CAD model).

**Figure 19 materials-17-00213-f019:**
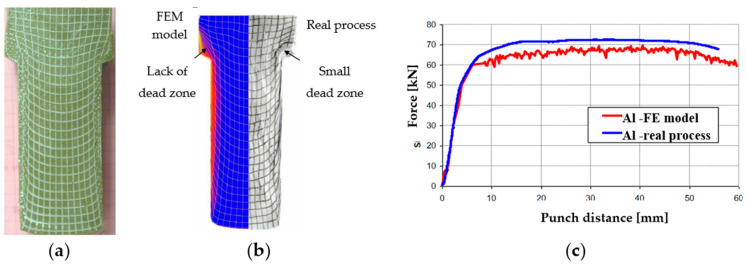
Research results: (**a**) the physical model, (**b**) the manner of material flow obtained from FEM (with the use of the special program option “flow line”) and the real process, (**c**) the course of the extrusion force in the function of the punch path for the real process and obtained from the numerical model after the friction coefficient correction.

**Figure 20 materials-17-00213-f020:**
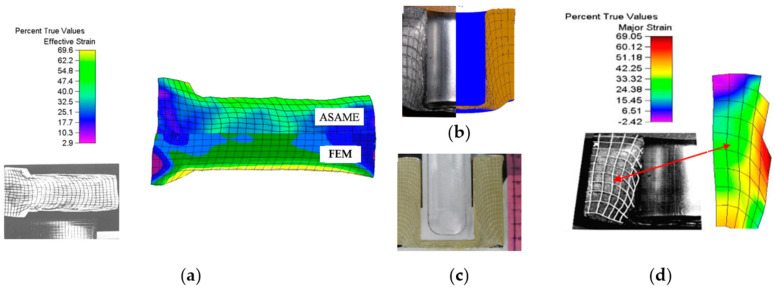
Results: (**a**) the equivalent deformation distributions obtained from FEM and determined by means of the ASAME program for the real process, (**b**) a comparison of the real process with the numerical model, (**c**) a physical model, (**d**) an analysis of deformations for Pb.

## Data Availability

Data are contained within the article.

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
