# Peer review of "Possibilities of Measuring and Detecting Defects of Forged Parts in Die Hot-Forging Processes"

_materials, 2023, doi:10.3390/ma17010213_

Round 1

Reviewer 1 Report

Comments and Suggestions for Authors

As is usual in this group of researchers, the title of the work is very long.

The novelty of the work is not clear.

What are its profound differences with respect to references 23 and 62?

The work presents 70 references of which 22 are self-citations, that is, 31%.

There are two works among the references in Polish (28 and 29). They must be eliminated because their usefulness is very restricted for other researchers.

57 of the 70 references are used in the introduction. There are practically no references that support the work carried out.

The paper is full of multiple reference types like (4,5) (6,7) (9,10,11) (12,13) ​​(17,18) (19,20,21,22).... This must be eliminated, leaving in each case only one reference that best represents the idea to which it is associated.

In relation to numerical modeling the authors could consider the following work by Metals (MDPI): Alcázar, J.; Abate, G.; Antunez, N.; Simoncelli, A.; Egea, A.J.S.; Krahmer, D.M.; López de Lacalle, N. Reduction of Die Wear and Structural Defects of Railway Screw Spike Heads Estimated by FEM. Metals 202111, 1834. https://doi.org/10.3390/met11111834. This paper shows how it is possible to simulate defects that occur in the real forging process.

The work is excessively long and it is strictly necessary to shorten it.

Author Response

Dear Reviewer,

Thank you for your insightful review and valuable comments, which we have responded to in a separate file.

regards,

Reviewer 2 Report

Comments and Suggestions for Authors

The paper presents research findings in the realm of die forging processes, primarily focusing on the utilization of advanced measurement techniques, tools, numerical modeling, and other IT methods for geometric analysis of forged items. The study outcomes are categorized into three main areas: 1st  area involves the application of tools such as optical scanners, along with related programs for data analysis., 2nd  area showcases results from measurements and analyses using FE modeling and specialized functions in calculation packages, addressing contact, flow lines, trap, or fold types to detect forging defects and analyze force parameters and 3rd area integrates various measurement and analysis methods, encompassing FEM, scanning, and other IT approaches for a comprehensive analysis of the geometry and defects of forgings.

Comments:

Materials and Methods

- In the concisely written chapter, only used devices are listed. This chapter does not contain research methodology.

- What is the measurement accuracy of the 3D scanner?

Results and Discussion

- The 1st paragraph in chapter 3.1 does not fit into the research results. More to the methodology.

- Figures 6b, 6d 7b are difficult to read, it is necessary to improve the quality of the images.

- What is the meaning of chapter 3.2 in the Results and Discussion section?

- line 275: "Based on the presented diagrams (Fig. 6ad), ...". I think, It should be Fig. 6b,d.

It is necessary to clearly describe novelty of this manuscript.

Comments on the Quality of English Language

English language (readibility) must be improved. The text contains very long and difficult to understand sentences.

Author Response

(The authors gave the same response as above.)

Round 2

Reviewer 1 Report

Comments and Suggestions for Authors

The new version of the work was improved significantly 

Author Response

Dear Reviewer,

Thank you for accepting our replies and once again for your valuable comments and suggestions.

Regards,

Reviewer 2 Report

Comments and Suggestions for Authors

1. Please, put the information about the accuracy of the 3D scanner in the manuscript. Not only as the answer to my question.

2. The novelty of this manuscript is well described, but only as the answer to my question. Put it in the manuscript as well.

Author Response

Dear Reviewer,

Thank you for accepting our replies and once again for your valuable comments and suggestions.

Of course, we took your two comments into account and added the relevant information in the revised manuscript.

Regards,